# Response-Related Factors of Bone Marrow-Derived Mesenchymal Stem Cells Transplantation in Patients with Alcoholic Cirrhosis

**DOI:** 10.3390/jcm8060862

**Published:** 2019-06-17

**Authors:** Haripriya Gupta, Gi Soo Youn, Sang Hak Han, Min Jea Shin, Sang Jun Yoon, Dae Hee Han, Na Young Lee, Dong Joon Kim, Soon Koo Baik, Ki Tae Suk

**Affiliations:** 1Institute for Liver and Digestive Diseases, Hallym University College of Medicine, Chuncheon 24253, Korea; phr.haripriya13@gmail.com (H.G.); gisu0428@gmail.com (G.S.Y.); wehome3@hallym.ac.kr (M.J.S.); ysjtlhuman@gmail.com (S.J.Y.); eogmlgks@gmail.com (D.H.H.); na0lee0218@gmail.com (N.Y.L.); djkim@hallym.ac.kr (D.J.K.); 2Department of Pathology, Hallym University College of Medicine, Chuncheon 24253, Korea; drhsh74@hallym.or.kr; 3Department of Internal Medicine, Wonju Severance Christian Hospital, Yonsei University, Wonju College of Medicine, Wonju 26426, Korea; baiksk@yonsei.ac.kr

**Keywords:** cirrhosis, bone marrow, mesenchymal stem cells, characteristics, liver regeneration

## Abstract

Liver cirrhosis leads to hepatic dysfunction and life-threatening conditions. Although the clinical efficacy of autologous bone marrow-derived mesenchymal stem cells (BM-MSC) transplantation in alcoholic cirrhosis (AC) was demonstrated, the relevant mechanism has not been elucidated. We aimed to identify the predictive factors and gene/pathways for responders after autologous BM-MSC transplantation. Fifty-five patients with biopsy-proven AC underwent autologous BM-MSC transplantation. The characteristics of responders who showed improvement in fibrosis score (≥1) after transplantation were compared with those of non-responders. BM-MSCs were analyzed with cDNA microarrays to identify gene/pathways that were differentially expressed in responders. Thirty-three patients (66%) were responders. A high initial Laennec score (*p* = 0.007, odds ratio 3.73) and performance of BM-MSC transplantation (*p* = 0.033, odds ratio 5.75) were predictive factors for responders. Three genes (olfactory receptor2L8, microRNA4520-2, and chloride intracellular channel protein3) were upregulated in responders, and CD36 and retinol-binding protein 4 are associated with the biologic processes that are dominant in non-responders. Eleven pathways (inositol phosphate, ATP-binding cassette transporters, protein-kinase signaling, extracellular matrix receptor interaction, endocytosis, phagosome, hematopoietic cell lineage, adipocytokine, peroxisome proliferator-activated receptor, fat digestion/absorption, and insulin resistance) were upregulated in non-responders (*p* < 0.05). BM-MSC transplantation may be warranted treatment for AC patients with high Laennec scores. Cell-based therapy utilizing response-related genes or pathways can be a treatment candidate.

## 1. Introduction

Cirrhosis is the end stage of liver injury with chronic necrosis and the inflammation of cells combined with fibrogenic processes, followed by the transformation of normal liver tissue into regenerative nodules [1,2]. Currently, the only definitive treatment for advanced liver cirrhosis is liver transplantation [3]. However, transplantation has limitations such as deficiency in donors, surgical complications, immunosuppression after transplantation, and high cost [4]. New therapeutic options, such as regeneration, bioartificial liver, and cell-based transplantation are currently under investigation to improve the prognosis of patients with liver cirrhosis [5].

With recent advancements and research-related cell therapies, mesenchymal stem cell (MSC) have demonstrated improved clinical applications because of their accessibility, expandability, and multipotentiality [6]. Various MSCs have been transplanted for the treatment of liver cirrhosis [7,8]. Recently, the positive influence of MSCs transplantation has been reaffirmed against alcoholic cirrhosis (AC), including clinical trials for autologous bone marrow-derived mesenchymal stem cell (BM-MSC) transplantation to patients with AC, which have shown an affirmative response of MSCs regarding reducing liver cirrhosis and improving liver function [9,10,11].

The therapeutic potentials of MSCs include their immunosuppressive properties, their ability to secrete various trophic factors, and hepatic regeneration [8,12]. After the homing of MSCs to injured tissue sites for repair, when in contact with local stimuli, such as inflammatory cytokines and ligands of toll-like receptors (TLRs), these stimuli can trigger MSCs to release a broad spectrum of bioactive molecules related to tissue remodeling and regeneration [10]. Since liver regeneration is mainly regulated by various cytokines and growth factors, such as insulin-like factor 1, fibroblast growth factors, hepatocyte growth factor, and chemokine ligand 2, they are potent mediators of angiogenesis and restrain cell apoptosis [13,14]. However, these mechanisms have been mostly investigated in preclinical studies, and application in clinical studies is still ambiguous.

In the current scenario, cell therapies, including hepatocytes or the direct infusion of cells in cirrhotic liver have been shown to be ineffective because obtaining human hepatocytes and maintaining their viability and function are difficult in clinical settings [10]. Despite the keen interest of using stem cells as opposed to liver transplantation, the therapeutic mechanisms have not yet been characterized [15,16]. To better understand the degree of responded and secreted factors by undifferentiated BM-MSCs, we tried to unravel the predictive factors for responders, response-related genes, and signal pathways after BM-MSC transplantation. In addition, we tried to explore the molecular, biological, and cellular functions of BM-MSCs in AC by microarray analysis so that this study may intercept novel insights into the gene-related therapeutic approaches in liver disease that can be utilized for further research.

## 2. Materials and Methods

### 2.1. Patients

From January 2013 to November 2015, a randomized open-label trial was prospectively conducted at 12 university-affiliated hospitals in Korea (clinicaltrial.gov: NCT01875081). Patients with biopsy-proven AC were recruited, and all the patients discontinued alcohol intake for at least six months before participation in this study. AC was diagnosed on the basis of imaging studies, history of alcohol consumption, and pathologic examinations of liver biopsy samples [17]. Abstaining from alcohol consumption for a six-month period has been shown to improve inflammation and facilitate stable results from liver histology, thereby increasing the power of the sample [18]. The eligibility criteria included age between 20–70 years and Child–Pugh scores of B or C. The institutional review boards of all the participating hospitals and the Korea Food and Drug Administration (KFDA) approved the protocol (KFDA clinical trial No. 27), and all the participating patients provided written informed consent to participate in this study. All the protocol used in the study conformed to the ethical guidelines of the 2013 Declaration of Helsinki.

We conducted a baseline evaluation, which included a family history, body mass index (BMI) calculation, abdominal ultrasound, a complete blood count, liver function test, and screening for viral markers. The serum biochemical parameters included total bilirubin, alanine aminotransferase, aspartate aminotransferase, gamma-glutamyl transpeptidase, alkaline phosphatase, albumin, blood urea nitrogen, creatinine, α-fetoprotein, prothrombin time, blood glucose, triglycerides, total protein, and total cholesterol. All the patients were tested for viral hepatitis, and their Child–Pugh scores, and model for end-stage liver disease (MELD) scores were calculated.

During the clinical trial, we collected BM-MSCs from passages 4–5 and media from subcultured samples. Patients were randomly assigned to a control group or one of two autologous BM-MSC groups, in which patients received one or two hepatic arterial injections of 5 × 10^7^ BM-MSCs 30 days after bone marrow aspiration.

Liver biopsies were performed twice, and biopsy specimens ≥15 mm (length) and ≥1.2 mm (width) were utilized for the analysis. Fibrosis status was measured with the Laennec and METAVIR scoring system by two liver pathologists who were blinded to the patients’ data. To calculate the chance-adjusted agreement, the kappa value was used for the interobserver agreement equaling the value of 0.947. When the two pathologists disagreed, they discussed and concluded the fibrosis score.

Patients in whom the fibrosis score (Laennec and METAVIR score) improved (by more than one point) post-transplantation were defined as responders. We compared BM-MSCs from passages 4–5 that showed significant good responses (*n* = 3) or no response (*n* = 3) after transplantation with BM-MSCs from the normal healthy control group (*n* = 3, age and gender matched).

In the analysis of BM-MSCs, cDNA microarrays and biomathematical analyses were used to identify the secreted genes and related pathways that were differentially expressed in specific stem cell populations in AC (Figure 1).

### 2.2. Bone Marrow Aspiration, Isolation, and Culture

All the manufacturing and product-testing procedures for the production of clinical-grade autologous MSCs (Livercellgram, Pharmicell Co., Ltd., Seongnam, Korea) were performed in a manner consistent with good manufacturing practices and the regulatory guidelines of the Ministry of Food and Drug Safety. With patients under local anesthesia, roughly 10–20 mL of bone marrow (BM) was aspirated from the posterior iliac crest.

Mononuclear BM cells were isolated using density-gradient centrifugation (Histopaque-1077, Sigma-Aldrich, St. Louis, MO, USA) and plated in 75-cm^2^ flasks (Falcon, Franklin Lakes, NJ, USA) with low-glucose Dulbecco’s modified Eagle’s medium (DMEM; Gibco, Grand Island, NY, USA) containing 10% fetal bovine serum (FBS, Gibco) and 20 g/mL gentamicin (Gibco), after which they were cultured at 37 °C in a 5% CO_2_ atmosphere. After 5–7 days, non-adherent cells were removed through replacement of the medium, and the adherent cells were cultured for 2–3 additional days. After detaching the colonized cells using a trypsin/EDTA (Ethylenediaminetetraacetic acid) solution (Gibco), they were re-plated in 175-cm^2^ flasks. When the cultures reached 70–80% confluence, serial subcultures were performed up to passages 4 or 5 for injection.

For the second injection in the two-injection BM-MSC group, some cells from passage 1 were harvested during the cell culture process and cryopreserved in 10% dimethyl sulfoxide (Sigma-Aldrich) and 90% FBS. These cells were thawed according to the injection schedule and subcultured to passage 4 or 5.

### 2.3. Viability, Immunophenotype, and Differential Potential of Stem Cells

A tryptan blue exclusion assay was used to assess the viability of the BM-MSCs. The immunophenotypes of the final harvested BM-MSCs were analyzed utilizing CD14, CD34, CD45, CD73, and CD105 as markers. To determine their immunophenotype, MSCs were stained with the following antibodies conjugated with phycoerythrin (PE) or fluorescein isothiocyanate (FITC): anti-CD14 FITC, anti-CD34 FITC, anti-CD45 FITC, anti-CD73 PE, and anti-CD105 PE (BD Biosciences, San Jose, CA, USA). Briefly, 5 × 10^5^ cells were resuspended in 0.2 mL of phosphate-buffered saline and incubated with antibodies at room temperature for 20 min. As the isotypes control, fluorescein isothiocyanate-conjugated or phycoerythrin-conjugated mouse IgGs (Immunoglobulin G) were used. The intensity of the fluorescence of the cells was evaluated using flow cytometry (NaviosTM; Beckman Coulter, Fullerton, CA, USA) (Figure 1).

Furthermore, the potential of BM-MSCs to differentiate into osteoblasts and adipocytes was tested following the procedures published in an earlier study [7]. Osteogenic differentiation was assessed by plating the cells at 2 × 10^4^ cells/cm^2^ in six-well plates and culturing them for 2–3 weeks in osteogenic medium composed of low-glucose DMEM medium supplemented with 10% FBS, 10 mM of β-glycerophosphate, 10^−7^ M of dexamethasone, and 0.2 mM of ascorbic acid (Sigma-Aldrich) [19]. To evaluate adipogenic differentiation, BM-MSCs were plated at 2 × 10^4^ cells/cm^2^ in six-well plates and cultured for 1 week. Differentiation was induced through an adipogenic medium composed of 10% FBS, 1 μM of dexamethasone, 0.5 mM of 3-isobutyl-1-methylxanthine, 10 μg/mL of insulin, and 100 μM of indomethacin in high-glucose DMEM for 3 additional weeks. The differentiated cells were fixed in 4% paraformaldehyde for 10 min, after which they were stained with fresh Oil Red-O solution (Sigma-Aldrich) to detect lipid droplets [7].

### 2.4. Whole Transcript Expression Arrays and Data Preparation

RNA extraction procedures were performed according to the manufacture’s guidelines. A ND-1000 Spectrophotometer (NanoDrop, Wilmington, DE, USA) and an Agilent 2100 Bioanalyzer (Agilent Technologies, Palo Alto, CA, USA) were used to evaluate purity and integrity. The Affymetrix Whole transcript expression array process was performed following the manufacturer’s protocol (GeneChip Whole Transcript PLUS reagent Kit, Thermo Fisher Scientific, MA, USA). cDNA was synthesized using the GeneChip WT (Whole Transcript) amplification kit, according to the manufacturer’s instructions. Fragmentation and biotin-labeling of the sense cDNA were performed with terminal deoxynucleotidyl transferase (TdT) using the GeneChip WT Terminal labeling kit. Roughy 5.5 μg of the labeled DNA target was hybridized to the Affymetrix GeneChip Human 2.0 ST Array at 45 °C for 16 h. The hybridized arrays were washed and stained using a GeneChip Fluidics Station 450 and scanned on a GCS3000 Scanner (Thermo Fisher Scientific, Waltham, MA, USA). The Affymetrix^®^ GeneChip™ Command Console software was used to compute signal values.

Raw data were automatically obtained by means of the Affymetrix data extraction protocol in the Affymetrix GeneChip^®^ Command Console^®^ software. Once the CEL files were imported, the data were summarized and normalized using the robust multi-average method implemented in the Affymetrix^®^ Expression Console™ software. The results of the gene-level analysis were exported and differentially expressed gene (DEG) analysis was performed.

A comparative analysis between the test and control samples was carried out using the LPE (local-pooled-error) test to calculate fold changes, and the null hypothesis was demonstrated and supported by the absence of any significant difference between the two groups. The false discovery rate was controlled by using the Benjamini–Hochberg algorithm to adjust the p-value. For each DEG set, a hierarchical cluster analysis was performed using complete linkage and Euclidean distance as a similarity measure.

### 2.5. Statistical Analysis

The chi-square test and Student’s t-test were used to evaluate baseline characteristics. Categorical variables were analyzed by the chi-square test, and continuous variables were assessed by Student’s t-test and paired t-test. Multivariate logistic regression was used to detect relating factors for responder BM-MSC transplantation. Data were analyzed with statistical software (SPSS, version 19.0, SPSS, Inc., Chicago, IL, USA) and GraphPad Prism version 6.0 for Windows (GraphPad Software, San Diego, CA, USA). For all the tests, *p*-values < 0.05 were considered significant.

Gene enrichment and functional annotation analyses for significant probe lists were performed using Gene Ontology (http://geneontology.org/) and KEGG (Kyoto encyclopedia of genes and genomes) (http://kegg.jp). All the statistical tests and visualization of differentially expressed genes were conducted using R v. 3.1.2. statistical software (www.r-project.org).

## 3. Results

### 3.1. Patients

A total of 72 patients with AC were randomly assigned to the three groups, and 17 patients were excluded after randomization. In the end, 55 patients (18 in the control group and 37 in the BM-MSC group) completed this study. After 6 months of the BM-MSC transplantation, liver biopsies were performed for the histological analysis of treated AC patients.

Thirty-three patients (66%) were identified as responders in this study. In the comparison between responders and non-responders, the initial high Laennec score was significantly increased only in the responder group (*p* < 0.05). Other variables did not show differences in baseline characteristics between responders and non-responders (Table 1 and Table 2).

In the multivariate analysis, the initial high Laennec score (*p* = 0.007, odds ratio 3.73) and performance of BM-MSC transplantation (*p* = 0.033, odds ratio 5.75) were predictive factors for the responders (Appendix A).

### 3.2. Viability, Immunophenotype, and Differentiation Potential

The mean viability percentage of the BM-MSCs, assessed by the trypan blue exclusion assay was approximately 86–89% in all the groups (approved viability is greater than 70%). In the immunophenotyping analysis, BM-MSCs were positive for CD73 and CD105 and negative for CD14, CD34, and CD45 [7]. At the second passage, it was observed that BM-MSCs had successfully differentiated into osteoblasts and adipocytes, as evidenced by 5-bromo-4-chloro-3-indolyl phosphate/intro blue tetrazolium staining for alkaline phosphatase activity and Oil Red-O staining for lipids.

### 3.3. Response-Related Biological Processes and Signal Pathway

In the analysis between the AC and control groups, a total of 1028 terms were identified. Among all the detected terms, 823 in biological processes, 109 in cellular components, and 98 in molecular functions were identified. The most significant terms are listed in Appendix A. In the analysis between non-responders and responders, all the related terms were divided into different processes, resulting in the identification of 1160 terms. Among all the detected terms, there were 947 in biological processes, 90 in cellular components, and 123 in molecular functions. The most significant of all the terms were found to be associated with biological processes, which are listed in Table 3.

Among all the evaluated pathways, 15 pathways showed statistical significance between the AC and normal control groups (Figure 2). The Ras signaling, Wnt signaling, antigen processing and presentation, Axon guidance, and graft-versus-host disease pathways were upregulated, whereas metabolic, arginine and proline metabolism, TGF (transforming growth factor)-beta signaling, HIF (hypoxia-inducible factor) signaling, FoxO signaling, cytokine interaction, stem cell pluripotency, natural killer cell-mediated cytotoxicity, and central carbon metabolism were all downregulated.

In the comparison between the non-responder and responder groups, 12 pathways showed statistical significance, which included the metabolic, inositol phosphate metabolism, ABC (ATP-binding cassette) transporters, AMPK (AMP-activated protein kinase) signaling, ECM (extracellular matrix) receptor interaction, endocytosis, phagosome, hematopoietic cell lineage, adipocytokine signaling, PPAR (peroxisome proliferator-activated receptor) signaling, fat digestion and absorption, olfactory transduction, and insulin resistance (Figure 3).

### 3.4. Differentially Expressed Genes

To detect the most dysregulated genes of significance, we showed the top 10 DEGs between the AC and normal control groups. Among the top 10 DEGs, eight DEGS (Glial cell-derived neurotrophic factor family receptor alpha-1 [GFRA1], fibrillin-2 precursor [FBN2], Ras P21 protein activator 4B [RASA4B], killer cell immunoglobulin like receptor, three immunoglobulin G domains, long cytoplasmic tail 2, and microRNA 4436a [KIR3DL2], and SERPINF1) were increased, and stanniocalcin-1 (STC1), RNA, U5F small nuclear 1 (RNU5F-1), tektin 4 pseudogene 2 (TEKT4P2), and zinc finger protein (ZFY) were downregulated in the AC group (Appendix A).

In the comparison the between non-responder and responder groups, 10 DEGs (maternally expressed gene 3 [MEG3], retinol binding protein 4 [RBP4], LOC101928395, sortilin related VPS10 domain containing receptor 2 [SORCS2], MIR2355, MIR520D, zinc finger protein Y-Linked platelet glycoprotein 4 [ZFY], chloride intracellular channel 3 [CLIC3], MIR4520-2 and olfactory receptor family 2 subfamily L member 8 [OR2L8]) were significantly upregulated, and CLIC3, MIR4520-2, and OR2L8 were downregulated in non-responders (Table 4).

## 4. Discussion

From our previous lab data, it was corroborated that the transplantation of BM-MSs to AC patients has proved to be new perspective in this area of research. An improved Child–Pugh score (*p* < 0.05) after 6 months of BM-MSC transplantation showed changes in liver fibrosis that provided insight regarding the regeneration of the liver tissue. Although other characteristics of the cirrhosis patients were statistically insignificant, mild improvement was observed, which provided valuable insight into the efficacy of BM-MSC transplantation in AC patients [9]. The related clinical factors and precise mechanism of response after BM-MSC transplantation have not been elucidated. Thus, this intriguing function of BM-MSCs led us to perform the microarray analysis so as to unfold possible genes and their signaling pathways involved in BM-MSCs function.

In this study, the initial high Laennec score (odds ratio 3.73) and BM-MSC transplantation (odds ratio 5.7) were predictive factors for the responders. In a previous in vivo study, MSC sheets prevented the growth of hepatocellular carcinoma and the progression of advanced stage of cirrhosis [20]. Regarding the initial high Laennec score, the exact mechanism of why the advanced stage in histology is prone to a good response is not clear. In this study, we strictly selected stable patients that abstained from alcohol consumption for 6 months. Since patients with a high Laennec score have increased the potential for the reversal of fibrosis, an initial high Laennec score might be supposed to be a predictor for good responders. Additional clinical studies on the mechanism are needed in the future.

In this study, 33 patients (66%) were identified as responders after transplantation, and MSC transplantation is another predictive factor for good responders. As we discussed in the previous trial, MSC transplantation has been identified as a promising treatment for liver regeneration in patients with AC [21,22]. As a result, BM-MSC transplantation effectively improved the fibrosis stage and liver testing in patients with AC.

In the present study, we identified specific genes and evaluated whether their pharmacological activity could affect the regulation of liver cirrhosis. By using a cDNA microarray and immunophenotyping assays, we focused on identifying MSC functions and related trophic factors for secreted genes and their pathways that are differentially expressed in specific stem cell populations in patients with liver cirrhosis.

Through microarray analysis, 466 DEGs were upregulated and 449 DEGs were downregulated in all the groups. This finding suggests that the effect of BM-MSC transplantation therapy in the AC group was a multigene-regulated complex process. In addition, the functions of the identified differentially expressed genes were involved in various biological processes, cellular components, and molecular functions and various related signaling pathways. Six genes were significantly upregulated, and nine genes were downregulated in the AC group compared with the control group. In the comparison between responders and non-responders, three genes (OR2L8, MIR4520–2, and CLIC3, *p* < 0.05) were functionally upregulated, and 28 genes are significantly downregulated in the responders.

Of biologic process and related genes, the lipid-associated process is upregulated, and CD36 is involved in most of the biologic processes that are dominant in non-responders. CD36 has also been implicated in hemostasis, thrombosis, malaria, inflammation, lipid metabolism, and atherogenesis [23,24]. CD36 is expressed on multiple cell types and has numerous functions. In a previous study, patients who have hematogenous stem cells transplantation with CD36 positive showed transfusion refractoriness, which was significantly improved after transplantation from CD36-deficienct donors [25]. Taken together, CD36 is related to non-responders after transplantation, and can be defined as a biomarker for expecting responsiveness in AC patients.

RBP4 is secreted mainly by hepatocytes and acts as the major transporter for vitamin A and retinol acid in serum [26,27]. RBP4 was overexpressed in non-responders in this study. The overexpression of RBP4 resulted in cancer cell migration, and RBP4 was upregulated in ovarian cancer cells. MMP-2 and MMP-9, which are key factors in cancer metastasis, were affected by RBP4 overexpression. This suggested that RBP4 stimulates cancer cell migration along with cell proliferation [28]. Overall, RBP4 could be a negative factor in the response to MSC transplantation, although it is related to cell migration in tumors.

CD36 and RBP4 are upregulated in non-responders and commonly associated genes in all biologic processes. CD36 and RBP4 can be used as predictive markers after transplantation, as the expression is negatively associated with poor outcomes in various human signal pathways. So far, the mechanism has not been elucidated, so further research studies evaluating the role of CD36 and RBP4 are needed in the future.

Long noncoding RNAs, such as the tumor-suppressing MEG3, have been identified as regulators of various types of human tumors. Previous reports have indicated a positive correlation with human diseases showing the abnormal differentiation of BM-MSCs [29]. In one report, MEG3 inhibits cell proliferation, and the downregulation of MEG3 activates the Wnt/beta-catenin signaling pathway via epigenetically regulating the Wnt genes promotor [30]. Another study demonstrated that the downregulation of MEG3 promotes BM-MSCs differentiation [31]. In this study, MEG3 expression was significantly increased in non-responders compared with responders. Taken together, MEG3 is downregulated in responders, and further studies using the MEG3 gene are needed in the future.

CLIC3 is clearly increased in the responder group in this result. CLIC3 acts as a key regulator of cell migration in tissues and as an independent prognostic indicator in pancreatic cancer [32]. Additionally, CLIC3 regulates the recycling of membrane-type 1 matrix metalloproteinase, and the transport of endosomes and/or lysosomes back to the plasma membrane, and demonstrates potential for the development of new therapies for more efficient mobilization and homing [33]. Moreover, CLIC3, which is expressed in mesenchymal stromal cells, demonstrated a lineage decision-making process, and is overexpressed during osteogenic differentiation [34]. This finding suggests that CLIC3 is a lineage-specific gene and could be a potential factor for the differentiation of MSCs to hepatocytes. Additional investigations should be performed to determine the exact mechanism of CLIC3 in the future.

Human miR-520d is a minor miRNA, and its upregulation was observed to induce tumor-suppressive effects and inhibit metastasis [35]. miR-520d-5p is suspected to act in the transformation of hepatoma cells to noncancerous cells in vivo via p53 upregulation [36]. The signaling pathway induced by miR-520d-5p has been shown to inhibit cell motility and invasiveness [37]. Another study demonstrated that miR-520d inhibits c-Myc, Cyclin D1, and MMP9 expression, and their inhibitory role demonstrated the reduction in the proliferation, invasion, and migration of cancer cells [38]. Our results are similar to those reported in previous studies. Additionally, non-responders showed an increased expression of miR-520d.

STC1, as a glycoprotein with pleiotropic effects, may show autocrine and paracrine functions. In two separate studies, STC1 mediates the anti-apoptotic effects of MSCs in lung epithelial cells [39], and MSCs exhibited anti-fibrotic impact via STC1 secretion in pulmonary fibrosis, which manifests paracrine function by the reduction in TGF-β1 production from alveolar macrophages [40]. Thus, these facts suggest that STC1 is directly related to MSCs, and the downregulation of STC1 disrupts the function of MSCs, which can be clearly seen in our study in the AC group. GFRA1 and FBN2 upregulation in the AC group can also be related with the previous studies that show implication in carcinogenesis [41,42].

The biological process analysis indicated that DEGs in the BM-MSC treated group were mainly involved in the processes of biological regulation, stress development, immune response, cellular localization, cellular response to hypoxia, and secretions. It has been found that HIF-1 regulates the expression of STC1 as well as BNIP3, and shows a relation to the Warburg effect in the progression of tumorigenesis under hypoxic conditions [43]. Both BNIP3 and STC1 are hypoxia-responsive genes, and their overexpression actuates tumor progression. Furthermore, the downregulation of these genes alleviates their effects and promotes cancer cell apoptosis. Additionally, MCSs under hypoxic conditions have been shown to downregulate the proapoptotic protein BNIP3 by exerting paracrine effects [44]. Metabolic reprogramming has been shown to increase tumorigenesis in harsh cellular environments, such as hypoxia and oxidative stress. A previous study also inferred that high expression in the liver of PDK1, a downstream target of HIF-1α factor, regulates metabolism by metabolic reprogramming, and it has been shown to promote liver metastasis [45]. Therefore, these studies, together with ours, specify the conclusive effect of BM-MSC transplantation in AC compared with the normal control group. BM-MSC treatment of AC patients was mediated by the interplay between multiple genes involved in the regulation of diversified biological systems.

Via KEGG pathway analysis, pathways that reduce liver injury and promote its growth were identified. BM-MSC transplantation upregulated the protein that negatively regulates the Ras signaling pathway and RASA4b. RASA4b is a GTPase-activating protein that subdues the RAS/MAPK pathway, resulting in the inactivation of the signaling pathway [46]. Since the aberrant activation of the RAS/MAPK signaling pathway is implicated in liver cancer progression [47], its inactivation can prevent downstream pathways and contribute to the survival of hepatocytes. However, RASA4b does not have a direct relationship with BM-MSCs, and further study is needed in order to establish a direct relationship with BM-MSCs. Thus, BM-MSC transplantation can improve cirrhosis in association with the participation of multiple genes and signaling pathways.

## 5. Conclusions

BM-MSC transplantation may be an effective treatment in patients with AC, and responders revealed different gene expression and signaling pathways such as CD36, RBP4, and lipid-associated processes compared with non-responders. Therefore, cell-based therapy utilizing response-related genes is warranted in patients with AC, and this study may act as a guide for future fundamental and clinical research.

## Figures and Tables

**Figure 1 jcm-08-00862-f001:**
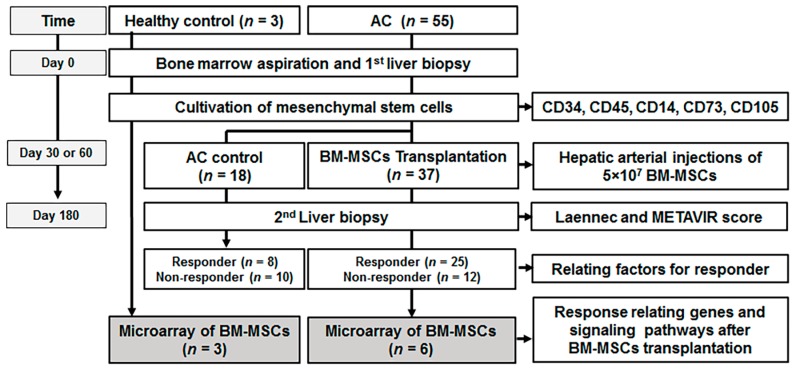
Study design. AC, alcoholic cirrhosis; BM-MSCs, bone marrow-derived mesenchymal stem cells; *n*, number.

**Figure 2 jcm-08-00862-f002:**
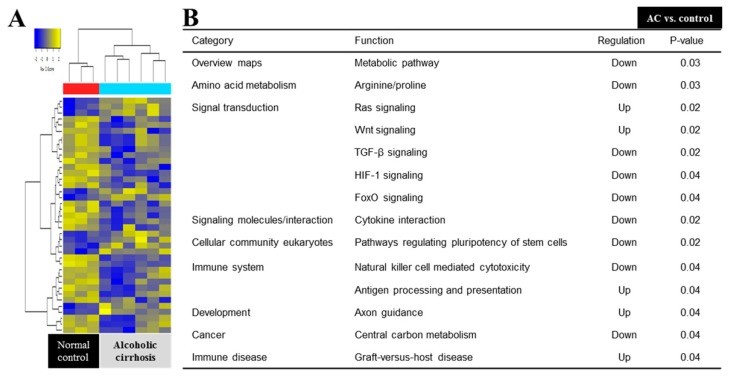
Microarray analysis between AC and normal control group. (**A**) Heat map depicting the expression of differentially expressed genes encoding various markers in BM-MSCs. (**B**) The KEGG (Kyoto encyclopedia of genes and genomes) pathway mapped for significantly dysregulated pathways in AC and the normal control group. TGF, transforming growth factor; HIF, hypoxia-inducible factor.

**Figure 3 jcm-08-00862-f003:**
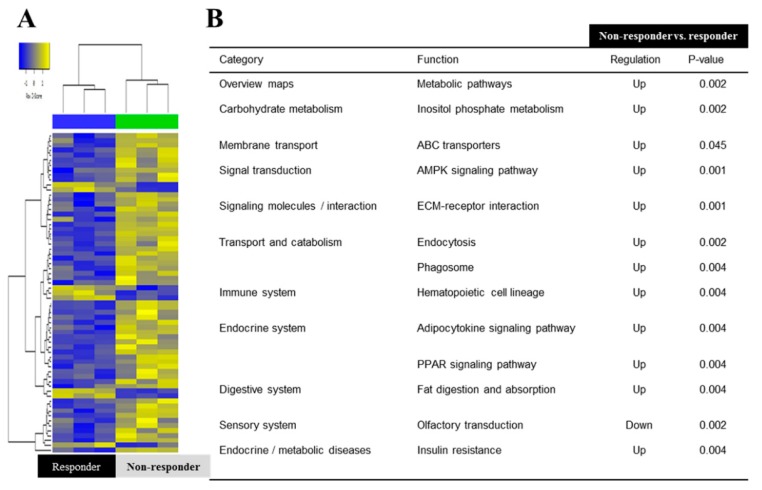
Microarray analysis between non-responders and responders. (**A**) Heat map depicting the expression of differentially expressed genes encoding various markers in BM-MSCs. (**B**) The KEGG Pathway mapped for significantly dysregulated pathways in non-responders and responders. ABC, ATP-binding cassette transporters; AMKP, AMP-activated protein kinase; ECM, extracellular matrix; PPAR, peroxisome proliferator-activated receptor.

**Table 1 jcm-08-00862-t001:** Baseline characteristics of patients.

Variables	AC Control				*p*	BM-MSC Transplantation				*p*
	Responder (*n* = 8)		Non-Responder (*n* = 10)			Responder (*n* = 25)		Non-Responder (*n* = 12)		
Male (*n*, %)	8	(100)	9	(90)	NS	21	(84)	9	(82)	NS
Age ^a^	51	(8)	54	(8)	NS	54	(8)	55	(9)	NS
Functional analysis ^a^										
CP score	8.5	(1.4.)	8.0	(1.5)	NS	7.8	(1.1)	7.8	(1.1)	NS
MELD score	14.9	(4.9)	12.8	(3.2)	NS	10.3	(3.9)	8.3	(4.7)	NS
Histologic analysis ^a^										
Laennec score	8.9	(1.4)	7.5	(1.1)	0.029	7.7	(1.2)	6.7	(1.2)	0.021
Biochemical analysis ^a^										
AST (IU/L)	61	(52)	42	(9)	NS	39	(18)	43	(15)	NS
ALT (IU/L)	37	(28)	28	(9)	NS	23	(10)	22	(13)	NS
Albumin (g/dL)	3.2	(0.6)	3.6	(0.5)	NS	3.7	(0.8)	3.3	(0.6)	NS
Bilirubin (mg/dL)	3.5	(2.9)	2.4	(1.5)	NS	1.6	(1.2)	1.5	(1.0)	NS
ALP (IU/L)	169	(66)	97	(35)	0.009	118	(68)	169	(88)	NS
GGT (IU/L)	74	(97)	61	(46)	NS	69	(68)	78	(53)	NS

^a^ Continuous variables are expressed as mean values (standard deviation) *n*, number; *p*, *p*-value; MELD, model for end-stage liver disease; AC, alcoholic cirrhosis; ALT, alanine aminotransferase; ALP, alkaline phosphatase; AST, aspartate aminotransferase; BM-MSC, bone marrow-derived mesenchymal stem cells; GGT, gamma glutamyl transferase; NS, not significant.

**Table 2 jcm-08-00862-t002:** Characteristics of responders and non-responders.

Variables	Responder (*n* = 33)				*p*	Non-Responder (*n* = 22)				*p*
	Pre		Post			Pre		Post		
Functional analysis ^a^										
Child–Pugh score	7.7	(1.1)	6.7	(1.7)	<0.001	8.1	(1.4)	7.3	(2.1)	NS
Histologic analysis ^a^										
Laennec score	8.0	(1.3)	6.3	(1.3)	<0.001	7.1	(1.2)	7.4	(1.5)	0.031
Biochemical analysis ^a^										
AST (IU/L)	44	(31)	38	(21)	0.012	47	(15)	49	(23)	0.036
ALT (IU/L)	25	(17)	21	(11)	NS	28	(12)	30	(14)	NS
Albumin (g/dL)	3.7	(0.7)	4.1	(1.9)	NS	3.3	(0.6)	3.3	(0.6)	NS
Bilirubin (mg/dL)	1.8	(1.2)	1.8	(1.5)	NS	2.7	(2.3)	2.1	(0.9)	NS
ALP (IU/L)	118	(59)	104	(35)	NS	142	(74)	126	(58)	NS
GGT (IU/L)	76	(70)	75	(50)	NS	57	(52)	77	(75)	NS

^a^ Continuous variables are expressed as mean values (standard deviation) *n*, number; MELD, AST, aspartate aminotransferase; ALT, alanine aminotransferase; ALP, alkaline phosphatase; GGT, gamma glutamyl transferase; NS, not significant.

**Table 3 jcm-08-00862-t003:** Significant biological process between non-responders and responders.

Term	*p*-Value	Genes
Lipid transport	0.012	ABCA9, RBP4, CD36
Lipid localization	0.015	ABCA9, RBP4, CD36
Secretion	0.022	MEG3, RBP4, STC1, CD36
System process	0.024	EYA4, OR2L8, MEG3, RBP4, STC1
Positive regulation: molecular mediator of immune response	0.037	RBP4, CD36
Establishment of localization	0.040	ABCA9, GEM, MEG3, RBP4, STC1, CLIC3, CD36
Anion transport	0.042	STC1, CLIC3, CD36

ABCA9, ATP binding cassette subfamily A member 9; RBP4, retinol binding protein-4; CD36, CD36 Molecule; MEG3, maternally expressed gene 3; STC1, stanniocalcin-1; EYA4, eyes absent homolog 4; OR2L8, olfactory receptor family 2 subfamily L member 8; GEM, GTP-binding protein GEM; CLIC3, chloride intracellular channel 3.

**Table 4 jcm-08-00862-t004:** Profiling of upregulated or downregulated gene expression in bone marrow-derived mesenchymal stem cells between the non-responder and responder groups.

Gene	Fold Change	Function
MEG3	2.93	Growth suppressor in tumor cells and cell growth activator for p53 and cell apoptosis
RBP4	2.38	Delivers retinol from the liver stores to the peripheral tissues
LOC101928395	2.22	NA
SORCS2	2.02	Represents the only module of the luminal/extracellular moiety Exhibit disparate functions depending on its proteolytic processing
LINC01111	1.97	NA
MIR2355	1.91	NA
MIR520D	1.86	Regulation of malignancy and maintaining p53 upregulation
ZFY	1.80	Function as a transcription factor
CD36	1.80	Function as receptors engaged in platelet adhesion to collagen Initiates signal transduction and internalization of receptor–ligand complexes.
STC1	1.79	Regulation of renal and intestinal calcium and phosphate transport, cell metabolism, or cellular calcium/phosphate homeostasis.
CLIC3	−2.02	Regulator of a recycling pathway and regulates cell migration and invasion
MIR4520–2	−1.70	NA
OR2L8	−1.61	Responsible for the recognition/G protein-mediated transduction of odorant signals.

MEG3, maternally expressed gene 3; RBP4, retinol binding protein 4; SORCS2, sortilin related VPS10 domain containing receptor 2; LINC01111, long intergenic non-protein coding RNA 1111; MIR, microRNA; ZFY, zinc finger protein Y-Linked platelet glycoprotein 4; STC, stanniocalcin1; CLIC3, chloride intracellular channel 3; OR2L8, olfactory receptor family 2 subfamily L member 8

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
