# Peer review of "Response-Related Factors of Bone Marrow-Derived Mesenchymal Stem Cells Transplantation in Patients with Alcoholic Cirrhosis"

_jcm, 2019, doi:10.3390/jcm8060862_

Round 1
Reviewer 1 Report
The authors state in Figure 1: 55 patients have been transplanted and out of them 33 were “responders”, while 22 were “non-responders”. So, that means 60% were “responders” and 40% - “non-responders”, but in line 174-175 of the manuscript: “In the end, 55 patients (18 in the control group and 37 in the BM-MSC group) completed this study. Thirty-three patients (66%) were identified as responders in this study.”
If 66% out of 55 patients, it would be 36.3 patients? If 66% out of 37, it would be 24.42 patients?
Moreover, seems like the control group belongs to this 55 patients’ group. In that case, 33/89.2% of patients out of 37 were “responders”. Does it mean that 4 “non-responders” and 18 controls were analyzed together as ‘non-responders” group? If so, the data provided do not properly reflect the results of the study.
The data of multivariate analysis has no meaning.
For readers to understand this study results I recommend clearly define the control group separately. The baseline characteristics of controls, “responders” and “non-responders” should be presented. The timetable of the transplantation procedures and follow up should be presented in details. In which time-point the “responders” achieved >1 improvement of fibrosis?
The authors should clearly motivate the selection of the genes for expression arrays analysis and discuss in details how their expression in BM-MSC is associated with autologous transplantation outcomes in alcoholic cirrhosis.
Author Response
jcm-513131
“Response relating factors of bone marrow-derived mesenchymal stem cells transplantation in patients with alcoholic cirrhosis”
Point-to-point responses to comments by the Reviewer 1
First of all, we would like to thank the Reviewer 1 for his/her comments, which helped us to improve this manuscript.
Reviewer 1
Comment 1: The authors state in Figure 1: 55 patients have been transplanted and out of them 33 were “responders”, while 22 were “non-responders”. So, that means 60% were “responders” and 40% - “non-responders”, but in line 174-175 of the manuscript: “In the end, 55 patients (18 in the control group and 37 in the BM-MSC group) completed this study. Thirty-three patients (66%) were identified as responders in this study.” If 66% out of 55 patients, it would be 36.3 patients? If 66% out of 37, it would be 24.42 patients? Moreover, seems like the control group belongs to this 55 patients’ group. In that case, 33/89.2% of patients out of 37 were “responders”. Does it mean that 4 “non-responders” and 18 controls were analyzed together as ‘non-responders” group? If so, the data provided do not properly reflect the results of the study.
group | TOTAL | ||||
Control AC | 1 BM-MSC | 2BM-MSC | |||
responder | NO | 10 | 5 | 7 | 22 |
YES | 8 | 13 | 12 | 33 | |
TOTAL | 18 | 18 | 19 | 55 | |
Response 1: We sincerely and deeply apologize for causing great confusion on the number of patients/group in the manuscript. We carefully reassessed the raw data and patients’ clinical findings and explained in detail in revised Figure 1.
In this clinical trial, all the 55 alcoholic cirrhosis patients were randomly assigned to three groups as mentioned in page 2 of 14 at line 88-90 “Patients were randomly assigned to three groups: one control group and two autologous BM-MSC groups involving one or two hepatic arterial injections of 5×107 BM-MSCs 30 days after bone marrow aspiration.” In the efficacy analysis between 1 hepatic arterial injection and two hepatic injection group, we cannot see the difference. Therefore, to avoid confusion and increase the number-induced power, we merged the both BM-MSC treated groups as one group and made the figure “Figure 1. Study design” and took healthy normal control BM-MSC samples (n=3) to compare with the BM-MSC treated group only. We made the following corrections to the ‘Figure 1. Study design’ section and made necessary changes to the manuscript. Moreover, we did not do microarray analysis of AC control group.
Original figure - attached file
Revised figure - attached file
Although AC control group (n=18) was given only best supportive treatment and abstinence from alcohol, 8 AC control group patients showed recovery and other patients (n=10) did not show response. To see the relating factors for the responder by using multivariate analysis, AC control group was included in analysis (revised Figure 1). Despite the keen interest of using stem cells as opposed to liver transplantation, the therapeutic mechanisms have not yet been characterized. So, we tried to show relating clinical factors, genes, and signal pathway of responder in this study.
Comment 2: The data of multivariate analysis has no meaning.
Response 2: Thank you for the suggestion. We have excluded Table 2 as reviewer recommended. With regard to multivariate analysis for relating clinical factors of responder, since BM-MSCs transplantation is an expensive and invasive procedure and clinicians do not know or expect response after BM-MSCs transplantations, we believe that it is necessary to provide information that can accurately predict the outcome. So, we kept multivariate analysis results in the paragraph.
“In the multivariate analysis, the initial high Laennec score (p=0.007, odds ratio 3.73) and performance of BM-MSC transplantation (p=0.033, odds ratio 5.75) were predictive factors for the responders.”
Comment 3: For readers to understand this study results I recommend clearly define the control group separately. The baseline characteristics of controls, “responders” and “non-responders” should be presented.
Response 3: This is a very important point that we totally agree with. We carefully compared biochemical and functional results of the groups (AC control vs. transplantation groups [responder and non-responder]). In our study, although AC control group (n=18) was given only best supportive treatment and abstinence from alcohol, 8 AC control group patients showed recovery and other patients (n=10) did not show response. To see the relating factors for the responder by using multivariate analysis, AC control group was included in the analysis (revised Figure 1). In more detail, we divided AC control group (responder and non-responder) and AC BM-MSC transplantation group (responder and non-responder) in the Table 1.
Revised Table 1.
Variables | AC control | P-value | BM-MSC transplantation | P-value | |||||||
Responder (n=8) | Non-responder (n=10) | Responder (n=25) | Non-responder (n=12) | ||||||||
Male (n, %) | 8 | (100) | 9 | (90) | NS | 21 | (84) | 9 | (82) | NS | |
Agea | 51 | (8) | 54 | (8) | NS | 54 | (8) | 55 | (9) | NS | |
Functional analysisa | |||||||||||
Child-Pugh score | 8.5 | (1.4.) | 8.0 | (1.5) | NS | 7.8 | (1.1) | 7.8 | (1.1) | NS | |
MELD score | 14.9 | (4.9) | 12.8 | (3.2) | NS | 10.3 | (3.9) | 8.3 | (4.7) | NS | |
Histologic analysisa | |||||||||||
Laennec score | 8.9 | (1.4) | 7.5 | (1.1) | 0.029 | 7.7 | (1.2) | 6.7 | (1.2) | 0.021 | |
Picrosirius red staining (%) | 8.9 | (1.4) | 7.7 | (1.3) | NS | 8.2 | (2.4) | 10.1 | (5.4) | NS | |
Biochemical analysisa | |||||||||||
AST (IU/L) | 61 | (52) | 42 | (9) | NS | 39 | (18) | 43 | (15) | NS | |
ALT (IU/L) | 37 | (28) | 28 | (9) | NS | 23 | (10) | 22 | (13) | NS | |
Albumin (g/dL) | 3.2 | (0.6) | 3.6 | (0.5) | NS | 3.7 | (0.8) | 3.3 | (0.6) | NS | |
Bilirubin (mg/dL) | 3.5 | (2.9) | 2.4 | (1.5) | NS | 1.6 | (1.2) | 1.5 | (1.0) | NS | |
ALP (IU/L) | 169 | (66) | 97 | (35) | 0.009 | 118 | (68) | 169 | (88) | NS | |
GGT (IU/L) | 74 | (97) | 61 | (46) | NS | 69 | (68) | 78 | (53) | NS | |
Comment 4: The timetable of the transplantation procedures and follow up should be presented in details. In which time-point the “responders” achieved >1 improvement of fibrosis?
Response 4: Thank you for your suggestion and we agree with the reviewer’s comment. When we designed this study, we tried to focus on the related genes that might be associated with the treatment of the BM-MSC transplantation in alcoholic cirrhosis patients. However, we missed to elaborate the time point of BM-MSC transplantation. We included that to the “Figure 1: Study design” present below. In addition, improvement of fibrosis can be seen by post Laennec scores which was significant (p=0.010).
Revised figure - attached file
Comment 5: The authors should clearly motivate the selection of the genes for expression arrays analysis and discuss in details how their expression in BM-MSC is associated with autologous transplantation outcomes in alcoholic cirrhosis.
Response 5: Thank you for highlighting this imperfection in our manuscript. We agree with your opinion that this manuscript should motivate the readers and provide a basic support in their future research. So for that we edited few lines in “Introduction” section and also added some lines 67-70 in “Introduction” section and below as well:
“In addition, we tried to explore molecular, biological and cellular functions of BM-MSCs in AC by microarray analysis so that this study may intercept novel insights into the gene related therapeutic approaches in liver disease that can be utilized for further research.”
Regarding details how their expression in BM-MSC is associated with autologous transplantation outcomes in alcoholic cirrhosis, we re-analyzed the microarray data and found that lipid associated process is dominantly activated in BM-MSCs of non-responder and RBP4 and CD36 are commonly associated genes in most biologic processes (below table, red color). RBP4 and CD36 are badly affecting genes in various human diseases such as transplantations, atherosclerosis, inflammation…. So, we mentioned their roles and our results in discussion section. (paragraph 6, 7, and 8).
Term | P-value | Genes |
Lipid transport | 0.012 | ABCA9, RBP4, CD36 |
Lipid localization | 0.015 | ABCA9, RBP4, CD36 |
Secretion | 0.022 | MEG3, RBP4, STC1, CD36 |
System process | 0.024 | EYA4, OR2L8, MEG3, RBP4, STC1 |
Positive regulation of production of molecular mediator of immune response | 0.037 | RBP4, CD36 |
Establishment of localization | 0.040 | ABCA9, GEM, MEG3, RBP4, STC1, CLIC3, CD36 |
Anion transport | 0.042 | STC1, CLIC3, CD36 |
Our results have a limitations in that we cannot clearly suggest the mechanism because we did not perform evidence proofing animal study. We suggested relating clinical findings, different genes and function, and different signal pathways. We mentioned this limitations on discussion section.
Reviewer 2 Report
The mechanisms of how and why BM-MSC can help in certain liver diseases have not been adequately described. In this study, the transplanted cells and possible differences between responding and non-responding recipients were examined.
The aim of this study and the experimental design are very interesting and worth exploring, but there are some incomprehensibilities within the manuscript. When comparing the responders and non-responders, the difference in the Laennic score appears to be the clearest factor, here. However, the functionality of the treatment with the cells is unfortunately not described here. The representation of the effectiveness of the cell transplantation (ie representation of the measured parameters before / after) is missing here. The parameters shown are describes as baseline data, but it is not clear at what timepoint these parameters were determined (e.g., how long after transplantation). These aspects were not sufficiently explained in the manuscript or are not fully comprehensible. The formulation of a clear objective might help the reader to understand this better. When reading the manuscript it seems as if the effect of the transplantation should be shown here, but the overall aim here is to investigate the mechanism or the regulation of relevant genes of the cells. The authors should edit the manuscript, so that the story will be more compact and defined for the reader, and thus a more comprehensible structure will be achieved.is
Author Response
jcm-513131
“Response relating factors of bone marrow-derived mesenchymal stem cells transplantation in patients with alcoholic cirrhosis”
Point-to-point responses to comments by the Reviewer 2
First of all, we would like to thank the Reviewer2 for his/her comments, which helped us to improve this manuscript.
Specific Comments:
Comment 1: The mechanisms of how and why BM-MSC can help in certain liver diseases have not been adequately described. In this study, the transplanted cells and possible differences between responding and non-responding recipients were examined.
Response 1: We appreciate the Reviewer’s thoughtful comment. We re-analyzed the microarray data and found that lipid associated process is dominantly activated in BM-MSCs of non-responder and RBP4 and CD36 are commonly associated genes in most biologic processes (below table, red color). RBP4 and CD36 are badly affecting genes in various human diseases such as transplantations, atherosclerosis, inflammation…. So, we mentioned their roles and our results in discussion section. (paragraph 6, 7, and 8)
Term | P-value | Genes |
Lipid transport | 0.012 | ABCA9, RBP4, CD36 |
Lipid localization | 0.015 | ABCA9, RBP4, CD36 |
Secretion | 0.022 | MEG3, RBP4, STC1, CD36 |
System process | 0.024 | EYA4, OR2L8, MEG3, RBP4, STC1 |
Positive regulation of production of molecular mediator of immune response | 0.037 | RBP4, CD36 |
Establishment of localization | 0.040 | ABCA9, GEM, MEG3, RBP4, STC1, CLIC3, CD36 |
Anion transport | 0.042 | STC1, CLIC3, CD36 |
Regarding “the mechanisms of how and why BM-MSC”, our results have a limitations in that we cannot clearly suggest the mechanism because we did not perform evidence proofing animal study. We suggested relating clinical findings, different genes and function, and different signal pathways. We mentioned this limitations on discussion section.
Comment 2: The aim of this study and the experimental design are very interesting and worth exploring, but there are some incomprehensibilities within the manuscript. When comparing the responders and non-responders, the difference in the Laennic score appears to be the clearest factor, here. However, the functionality of the treatment with the cells is unfortunately not described here. The representation of the effectiveness of the cell transplantation (ie representation of the measured parameters before / after) is missing here. The parameters shown are describes as baseline data, but it is not clear at what timepoint these parameters were determined (e.g., how long after transplantation). These aspects were not sufficiently explained in the manuscript or are not fully comprehensible. The formulation of a clear objective might help the reader to understand this better. When reading the manuscript it seems as if the effect of the transplantation should be shown here, but the overall aim here is to investigate the mechanism or the regulation of relevant genes of the cells. The authors should edit the manuscript, so that the story will be more compact and defined for the reader, and thus a more comprehensible structure will be achieved.
Response 2: Thank you for your valuable suggestions. We added Table 2 with functional data (pre and post BM-MSCs transplantation)
Table 2. Characteristics of responder and non-responder
Variables | Responder (n=33) | P-value | Non-responder (n=22) | P-value | |||||||
Pre | Post | Pre | Post | ||||||||
Functional analysisa | |||||||||||
Child-Pugh score | 7.7 | (1.1) | 6.7 | (1.7) | <0.001< span=""> | 8.1 | (1.4) | 7.3 | (2.1) | NS | |
Histologic analysisa | |||||||||||
Laennec score | 8.0 | (1.3) | 6.3 | (1.3) | <0.001< span=""> | 7.1 | (1.2) | 7.4 | (1.5) | 0.031 | |
Biochemical analysisa | |||||||||||
AST (IU/L) | 44 | (31) | 38 | (21) | 0.012 | 47 | (15) | 49 | (23) | 0.036 | |
ALT (IU/L) | 25 | (17) | 21 | (11) | NS | 28 | (12) | 30 | (14) | NS | |
Albumin (g/dL) | 3.7 | (0.7) | 4.1 | (1.9) | NS | 3.3 | (0.6) | 3.3 | (0.6) | NS | |
Bilirubin (mg/dL) | 1.8 | (1.2) | 1.8 | (1.5) | NS | 2.7 | (2.3) | 2.1 | (0.9) | NS | |
ALP (IU/L) | 118 | (59) | 104 | (35) | NS | 142 | (74) | 126 | (58) | NS | |
GGT (IU/L) | 76 | (70) | 75 | (50) | NS | 57 | (52) | 77 | (75) | NS | |
a Continuous variables are expressed as mean values (standard deviation)
n, number; MELD, AST, aspartate aminotransferase; ALT, alanine aminotransferase; ALP, alkaline phosphatase; GGT, gamma glutamyl transferase; NS, not significant
This manuscript is extension of our previous work and that data is already published. In this manuscript, we tried to describe and accumulate the possible factors and characteristics for AC patients and their effective treatment of BM-MSC transplantation. So, we majorly explored in this manuscript about possible genes associated with liver regeneration and related signaling mechanism involved with respect to differentially expressed genes. This manuscript provided a clear insight of the related factors and genes which is worth exploring in near future by researchers in this area of study. And would be very helpful to elucidate more and accurate functions of the possible involved genes that may portends lack of efficacy in treatments of liver diseases.
However, as per you suggestion, we edited and added few lines in “Result” section which is not shown here. Also added a paragraph under “Discussion” section from line 272 to 280 and mentioned below as well:
“From our previous lab data, it was corroborated that transplantation of BM-MSs to alcoholic cirrhosis patients has proved to be new perspective in this area of research. Improved Child-Pugh score (p<0.05) after 6 months of BM-MSC transplantation showed changes in the liver fibrosis which provided insight of regeneration of the liver tissue. Although other characteristics of the cirrhosis patients were statistically insignificant, but mild improvement was observed which provided valuable insight into the efficacy of BM-MSC transplantation in alcoholic cirrhosis patients [9]. The related clinical factors and precise mechanism of response after BM-MSC transplantation have not been elucidated. Thus, this intriguing function of BM-MSCs led us to perform the microarray analysis so as to unfold possible genes and their signaling mechanism involved in BM-MSCs function.”
Round 2
Reviewer 1 Report
1. In the abstract of the revised manuscript (lines 22-23) and in Results section authors state: “The characteristics of responders who showed improvement in Laennec score (≥ 1) after transplantation were compared with those of non-responders.
While in the Method section (lines 97-98) they write: “Patients in whom the METAVIR fibrosis score improved ( by more than 1 point) post-transplantation were defined as responders”.
What sort of improvement was achieved if any in the „responders“? Authors should take into account the bias of liver biopsy procedure: depending on the place of biopsy the same liver can exhibit different histological views.
2. I would recommend describing the normal healthy control group (n=3). Did those subjects match the patients by age/gender?
3. The conclusions of this study should be more careful and subtle.
The following conclusions are unsubstantiated by the study results:
“BM-MSC transplantation is warranted treatment for AC patients with high Laennec score”
“BM-MSC transplantation is an effective treatment in patients with AC”,
Author Response
Comment 1: In the abstract of the revised manuscript (lines 22-23) and in Results section authors state: “The characteristics of responders who showed improvement in Laennec score (≥ 1) after transplantation were compared with those of non-responders. While in the Method section (lines 97-98) they write: “Patients in whom the METAVIR fibrosis score improved ( by more than 1 point) post-transplantation were defined as responders”. What sort of improvement was achieved if any in the „responders“? C sc
Response 1: We appreciate your suggestions and we apologize for the confusion regarding fibrosis score. We sub-classified patients’ fibrosis according to METAVIR and Laennec score. Because Laennec fibrosis score is developed for the classification of METAVIR Fibrosis 4 (Laennec Aà Laennec Bà Laennec C), clinicians can see the fibrosis grades sequentially and we can grade fibrosis by numbers. We changed “Laennec score (≥ 1)” to “fibrosis score” on Abstract (lines 22-23) and we changed “METAVIR fibrosis score improved (by more than 1 point)” to “fibrosis score (Laennec and METAVIR score) improved (by more than 1 point)” on Method. This change was already mentioned in the “Figure 1”.
Comment 2: Authors should take into account the bias of liver biopsy procedure: depending on the place of biopsy the same liver can exhibit different histological views.
Response 2: Thanks for raising this point. We mentioned liver biopsy and Kappa value in the Method section.
“Two times of liver biopsies were performed and biopsy specimens with ≥ 15 mm (length) and ≥ 1.2 mm (width) were utilized for the analysis. Fibrosis status was measured with the Laennec and METAVIR scoring system by two liver-pathologists who were blinded to patients’ data. To calculate the chance-adjusted agreement, the kappa value was used for the inter-observer agreement equaling the value of 0.947. When the two pathologists disagreed, they discussed and concluded fibrosis score.”
Comment 3: I would recommend describing the normal healthy control group (n=3). Did those subjects match the patients by age/gender?
Response 3: We appreciate the Reviewer’s thoughtful comment. We selected “healthy control” with age and gender match control. We mentioned this on method section.
We compared BM-MSCs from passages 4-5 that showed significant good responses (n=3) or no response (n=3) after transplantation with BM-MSCs from the normal healthy control group (n=3, age and gender matched).
Comment 4: The conclusions of this study should be more careful and subtle. The following conclusions are unsubstantiated by the study results: “BM-MSC transplantation is warranted treatment for AC patients with high Laennec score” “BM-MSC transplantation is an effective treatment in patients with AC”,
Response 4: Thank you for the valuable comment. Efficacy of this treatment was considered on the basis of Laennac score which is showed significant improvement in AC patients. And, from our study we can conclude that BM-MSC may be future effective treatment for AC patients. Although this is not approved treatment yet, we made some minor changes in in “Conclusion section”.
“BM-MSC transplantation may be warranted treatment for AC patients with high Laennec score.”
“BM-MSC transplantation may be an effective treatment in patients with AC, and responders revealed different gene expression and signaling pathways such as CD36, RBP4, and lipid associated process compared with non-responders.”

Reviewer 2 Report
The authors have dealt constructively with the criticized points and edited the ambiguities. By editing the manuscript the story is now easier to understand.
Author Response
Thanks for yur kind review.
